# 3D PixBrush: Image-Guided Local Texture Synthesis

## Abstract

We present 3D PixBrush, a method for performing image-driven edits of local regions on 3D meshes. 3D PixBrush predicts a localization mask and a synthesized texture that are guided by the object in the reference image. Our predicted localizations are both *globally* coherent and *locally* precise. Globally - our method contextualizes the object in the reference image and automatically positions it onto the input mesh. Locally - our method produces masks that conform to the geometry of the reference image. Notably, our method does not require any user input (in the form of scribbles or bounding boxes) to achieve accurate localizations. Instead, our method predicts a localization mask on the 3D mesh from scratch. To achieve this, we propose a modification to the score distillation sampling technique which incorporates both the predicted localization and the reference image, referred to as localization-modulated image guidance. We demonstrate the effectiveness of our proposed technique on a wide variety of meshes and images.

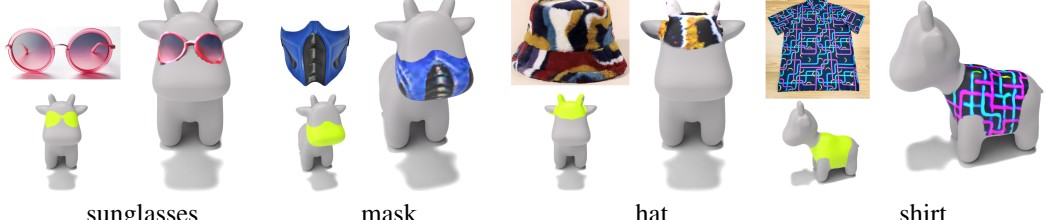

sunglasses        mask        hat        shirt

Figure 1: **3D PixBrush** produces localized textures on meshes driven by a reference image. We use a text prompt to initialize the localization and refine it to match the reference image. 3D PixBrush predicts a localization mask (yellow) and a texture that capture how the object in the reference image could plausibly be synthesized on the shape.

## 1 Introduction

When 3D artists manually texture local regions on meshes, they often use a reference image as guidance. However, it's challenging to create automated tools that mimic this approach: it requires using a reference image both to accurately localize a fine-grained segmentation and to texture that region so that it adheres to the style of the reference image. A promising strategy is to leverage the recent emergence of text-to-image (T2I) models Rombach et al. (2022); Ho et al. (2020); Saharia et al. (2022) for guiding local texturing of 3D shapes, which has been shown to work well when using text prompts as input Decatur et al. (2024); Sella et al. (2023); Zhuang et al. (2023). While intuitive, such text prompts lack precision regarding the attribute's appearance and structure. For example in Fig. 1, it is difficult to describe the pattern in the neon 'shirt' result or the shape of the segmentation in the blue 'mask' result.

Recent advances in T2I models have enabled images to act as prompts, providing a new level of control by guiding image generation with visual references Ye et al. (2023); Zhang et al. (2023). Recently, EASI-Tex Perla et al. (2024) presented a technique for *globally* stylizing 3D shapes using images. However, EASI-Tex cannot make *local* stylizations. Other works such TIP-Editor Zhuang et al. (2024) and Focal Dreamer Li et al. (2023b) perform local edits to 3D representations, but rely on manual user input to specify the localization region. In this work, we propose a technique to

Figure 2: **Localization Modulated Guidance.** Our method (right) uses text, image, and a learned (continually updating) mask to achieve a localized texture which adheres to the reference image. Using only text as input (left), yields a localized result that does not capture all the details of the reference image. Using text and image as input without a continually updating mask (middle), results in textures that resemble the reference image but are not properly localized.

enable using a reference image to synthesize a *localized* texture (*i.e.*, predict an explicit localization mask and a texture) on a 3D shape. This task requires understanding the *global* context of the reference image, *i.e. where* to position the reference image on the shape. Further, it requires *local* fine-grained understanding, *i.e.* synthesizing textures in 3D that adhere to the structure and style of the reference image. Additionally, our task is particularly challenging as explicitly blending local textures onto bare meshes prominently showcases any artifacts in the underlying segmentation mask.

We present 3D PixBrush, a method for using a *reference image* to prescribe a *local texture* to a 3D mesh. Our method can capture the object in the reference image and faithfully synthesize it on the 3D shape. Notably, our method accurately localizes a segmentation region on the surface of the 3D shape which is driven by the object in the reference image, as well as a text prompt. This enables our method to produce localized textures that capture the shape of the object in the reference image, beyond what can be generically localized with text alone (see Fig. 5). Moreover, our method synthesizes textures that both capture the style of the reference image, and conform to the predicted localization region. As far as we are aware, no method on *any* 3D representation (*e.g.* meshes, nerfs, gaussian splats) can use a reference image to both predict an explicit fine-grained segmentation in 3D and create a corresponding local texture contained to that region.

A key innovation of our work is the ability to predict a localization mask which accurately *contextualizes* the reference image onto the mesh surface (yellow insets in Fig. 1). Our predicted segmentation masks are both *globally coherent* and *locally precise*. Globally - our predicted masks are automatically positioned appropriately on the mesh surface (*e.g.* the eye glasses appear in the eye region in Fig. 1). Locally - our predicted masks contain the precise geometric structure of the reference image (*e.g.* the round eye glass shape in Fig. 1). Notably, our method is able to predict such segmentation masks without any user assistance (*e.g.* in the form of scribbles or bounding boxes). In tandem, we predict textures that, within the localized region of our predicted mask, accurately capture the content of the reference image (note the coloring of the sunglasses in Fig. 1).

We synthesize a localization mask and texture directly on the input mesh using a modified version of score distillation sampling (SDS) Poole et al. (2023); Wang et al. (2023). A straightforward application of SDS to optimize localized textures using only text guidance does not take into account the reference image (see Fig. 2, left). Notably, text does not capture the intricate details, textures, and spatial configurations depicted by an image. To incorporate a reference image as guidance, we leverage an image conditioned IP-Adapter model Ye et al. (2023). However, simply using IP Adapter for image guidance will generate textures that are aligned to the guiding image, but are not properly localized on the input shape (Fig. 2, middle). We propose a technique that enables combining the desirable properties of text and images to enable synthesizing textures that are both image-guided and explicitly localized, referred to as *localization modulated image guidance* (LMIG). Using LMIG results in synthesized textures that adhere to the input image while being accurately localized (Fig. 2, right). LMIG achieves this by modulating the cross-attention features of the image guidance with a continually updating mask. We use our currently predicted localization mask to perform *explicit* masking (*i.e.* foreground matting), and *attention*-based masking (LMIG). This enables precisely capturing the structure of the reference image by utilizing the continually updated localization directly within our system in order to strengthen our supervision. As a result, our localizations precisely reflect the structure of the reference image, leading to segmentations that are far more specific and detailed than those obtained through text alone (see Fig. 5).

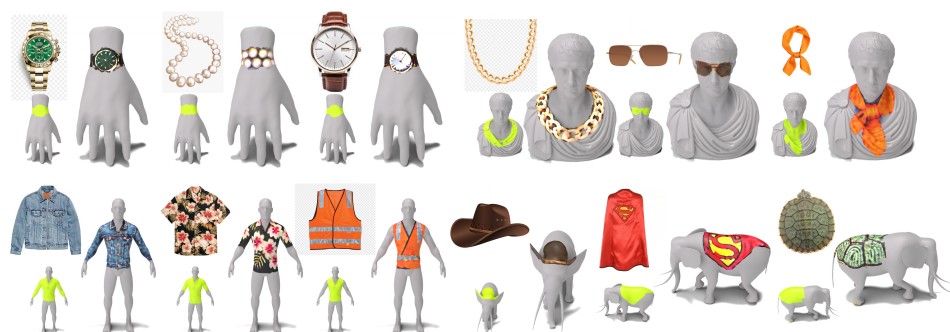

Figure 3: **Gallery.** We perform image-driven texturing of local regions on 3D meshes, using a reference image to both localize the relevant region on the mesh and simultaneously learn a texture that captures the style of the reference image. Our predicted local textures respect the global understanding of the shape and capture the fine-grained details of the reference image.

In summary, we present a method to create local texture edits on meshes guided by images. Our method synthesizes local textures on the shape surface such that they match the style of the reference image. Our method is the first to produce image-guided local textures and corresponding localization masks on meshes without any manual user input. We propose a localization modulation technique for our image guidance which allows us to coherently integrate local textures onto the input mesh. Our LMIG produces local textures that respect the global understanding of the shape, and capture the fine-grained details of the reference image. We showcase the effectiveness of our approach on a wide variety of meshes and reference images.

## 2 RELATED WORK

A large body of existing research employs neural networks to generate and refine 3D content. Some use explicit data structures such as meshes Michel et al. (2022); Oechsle et al. (2019); Bokhovkin et al. (2023); Siddiqui et al. (2022); Huang et al. (2020); Wei et al. (2021); Mohammad Khalid et al. (2022); Metzer et al. (2023); Ma et al. (2023); Lei et al. (2022); Hertz et al. (2020); Liu et al. (2020); Yin et al. (2021); Höllein et al. (2022); Gao et al. (2021) while others use implicit representations such as NeRFs Mildenhall et al. (2020); Zhang et al. (2022); Liu et al. (2023a); Fan et al. (2022) or 3D Gaussian Splats Kerbl et al. (2023); Tang et al. (2023). These works largely focus on generating 3D content from scratch. In contrast, some approaches focus on editing existing 3D content Gao et al. (2023); Kim et al. (2024); Chen et al. (2024); Haque et al. (2023), however, they apply edits globally instead of altering only local components. Additionally, the majority of these works use large, pre-trained text-to-image models Ho et al. (2020; 2022); Rombach et al. (2022) for supervision, which enable text guidance Lukoianov et al. (2024); Poole et al. (2023); Shi et al. (2023); Wang et al. (2023), but do not support generation or editing conditioned on images. Improved text-to-image models now support additional image guidance Ye et al. (2023); Zhang et al. (2023) and these models have been used to address this limitation Perla et al. (2024). However, image guided approaches still struggle to restrict their edits to local regions.

**Text-guided 3D Generation and Editing.** Large, pre-trained text-based diffusion models have been widely used by existing approaches to generate and edit 3D content. Score distillation (SDS) Poole et al. (2023); Wang et al. (2023) and its variants Lukoianov et al. (2024); Decatur et al. (2024); Katzir et al. (2023) are most commonly used to supervise 3D methods with 2D models. Many works use SDS for text guided 3D generation of both geometry and style from scratch Poole et al. (2023); Wang et al. (2023); Zhu & Zhuang (2023); Tsalicoglou et al. (2023); Lin et al. (2023); Chen et al. (2023b); Shi et al. (2023); Tsalicoglou et al. (2023); Katzir et al. (2023), while other focus on the stylization of existing geometry Metzer et al. (2023); Richardson et al. (2023); Chen et al. (2023a); Michel et al. (2022). These methods can be used to perform edits to existing 3D content Ruiz et al. (2023); Zhuang et al. (2023); Haque et al. (2023); Chen et al. (2024); Wang et al. (2024), but the resulting edits are global since they do not explicitly learn a localization region in which to contain their changes. Our work focuses on local editing which requires explicit segmentation of the localization region.

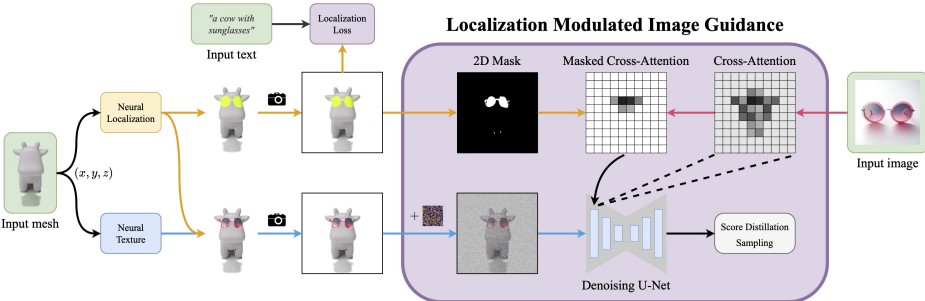

Figure 4: **Overview of 3D PixBrush.** The input mesh passes through a neural localization network that predicts a mask for the local texture edit region, and a Neural Texture network that generates a texture map for the shape. The networks are guided by a reference input image and a text prompt. The predicted localization is used to explicitly mask the predicted textures (curved orange arrow). The localization is also used to mask the cross-attention between the reference image and the activations from a rendering of the edited mesh. While the text prompt does not contain *any information* about the structure or color of the edit, our method properly captures the fine-grained details of the reference object depicted in the image.

**Text-guided Localization and Local Editing.** In order to ensure that edits are only applied locally, recent methods have looked to incorporate explicit localizations into their editing pipeline. However, obtaining precise localizations is challenging. Some works Li et al. (2023b); Zhuang et al. (2024); Sabat et al. (2024) opt to take these localizations as input from the user which results in accurate localizations, but comes at the cost of additional manual input. Another line of work aims to learn explicit localizations on 3D shapes using only text guidance Decatur et al. (2023); Abdelreheem et al. (2023b;a); Zhong et al. (2024). While these are able to produce localizations, they do not produce an accompanying edit and the task of matching up independently generated localizations and edits is non-trivial. 3D Paintbrush Decatur et al. (2024) and DreamEditor Zhuang et al. (2023) combine text-driven localization and text-driven editing to produce local edits confined to the desired region. However, these approaches can only be guided by text which is limited in its ability to precisely describe the details of the desired edit (see Fig. 7).

**Image Guidance for 3D Generation and Editing.** Using image-conditioned models Ye et al. (2023); Liu et al. (2023c), recent methods have found success at generating and editing 3D content with reference images Gao et al. (2022); Liu et al. (2023b); Qian et al. (2023); Liu et al. (2023c); Perla et al. (2024); Sabat et al. (2024); Zhuang et al. (2024). Many image-driven approaches operate on implicit representations Sabat et al. (2024); Zhuang et al. (2024), while EASI-Tex Perla et al. (2024) uses IP-Adapter Ye et al. (2023) guidance to generate a global texture over a mesh. Yet, just as with the text-driven methods, without explicit localization, precise local editing is very difficult. This is made even more challenging by the fact that existing methods do not exist for performing image-driven localization as they do with text. Nerf-Insert Sabat et al. (2024) and TIP-Editor Zhuang et al. (2024) both rely on user-defined localization regions in order to ensure their edits are applied locally. In contrast, our method aims to introduce image-driven localization alongside image-driven editing into a single automated process, producing accurate localizations and precise local texture edits on 3D meshes using image guidance.

## 3 METHOD

An overview of our method 3D PixBrush is illustrated in Fig. 4. The input to our system is a 3D mesh, a reference image $I$ that precisely defines our desired edit, and a text prompt, $y$, that describes the type of object in image (to convey semantic information important for localization) but does not contain information about the appearance of the object in the image. Our approach produces localizations and corresponding local textures over the 3D shape, both of which capture the nuanced details of the reference image $I$.

Our optimization is supervised using score distillation losses with pretrained diffusion models. For our localization loss, we use a text-to-image model and for our image guidance, we use an IP-Adapter Ye et al. (2023) which provides global visual supervision to the generated image. To enable

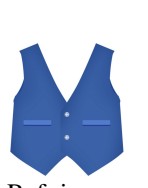 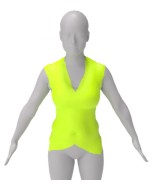 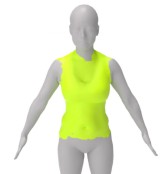 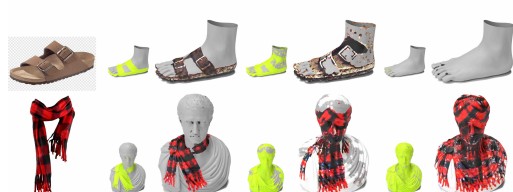

Ref. image      Ours      Text only: 'vest'

Figure 5: **Importance of Image Guidance on Localization**. Using a reference image of a blue vest (left), 3D PixBrush (Ours) produces an accurate localization that precisely captures specific structural details such as the sharp v neck and points at the bottom in the blue vest. If we produce a localization using only text guidance, the localization has no influence from the reference vest image and produces a generic vest as that misses many of the features in the reference vest.

Ref. image      Ours      w/o CA mask    w/o warm up

Figure 6: **Ablations**. 3D PixBrush (Ours) produces accurate localizations and detailed textures. Removing the localization modulated cross attention masking from the image guidance (w/o CA mask) leads to poor localizations and textures that capture some details, but are lower quality and do not respect the global context of the shape. Removing the initial localization loss (w/o warm up) often results in localizations that include the entire mesh and global textures that ignore geometry.

local supervision from the reference image $I$, we propose Localization Modulated Image Guidance (LMIG), a technique that leverages our current localization prediction to contextualize the image supervision for our local texture edit during the score distillation process.

### 3.1 NEURAL LOCALIZATION AND TEXTURING

3D PixBrush predicts localizations and local textures with neural fields defined over the mesh surface. We employ two networks, a neural localization network $F_\theta$ mapping points $x \in R^3$ to probabilities $p \in [0, 1]$, and a neural texture network $F_\phi$ mapping points $x \in R^3$ to RGB colors $c \in [0, 1]^3$. In practice, we represent these neural fields with Multi-Layer Perceptrons (MLPs) due to their bias towards smoothness Rahaman et al. (2019) which reduces speckling artifacts. However, we still want the ability to output high-frequency localizations and textures so we incorporate positional encoding layers Tancik et al. (2020) into our MLPs. Specifically, each network consists of a positional encoding layer followed by 6 blocks comprised of a fully connected layer, batch normalization layer, and ReLU activation.

To optimize our neural localization and texture, we invert a UV parametrization of our mesh to obtain query points on the 3D surface that correspond to the centers of texel coordinates on a texture plane. These points are then passed through the MLPs to obtain predicted localizations and textures over the mesh surface. We can trivially extract texture maps from the neural fields allowing for seamless rendering and integration into existing graphics pipelines.

### 3.2 COMBINED TEXT AND IMAGE GUIDANCE

We supervise our method using score distillation from pretrained 2D models. In each iteration of our optimization, our networks predict probabilities and texture values over the surface of the mesh, so we need a way to visualize these predictions in 2D in order to supervise using score distillation from a 2D model. For our localizations, we follow 3D Highlighter Decatur et al. (2023), directly visualizing the predicted probabilities as a texture and rendering that.

To visualize our local textures, we follow 3D Paintbrush Decatur et al. (2024) and use our predicted localization probabilities to mask our texture prediction to just the region we are interested in, giving us a local texture which we then render as well. The explicit mask of our texture with the current prediction of the localization allows for gradients from our texture renders to flow back to our localization. This enables us to update our localization using losses (specifically, our image-guided loss) applied to renders of our local texture.

Our optimization is guided by two losses: a text-driven loss applied to our localization renders and an image-driven loss applied to our texture renders. The image-guided loss is primarily responsible for learning a texture to match the reference image. Yet, due to the explicit masking of our textures with the predicted localizations (as described above), the image guidance loss applied solely to our

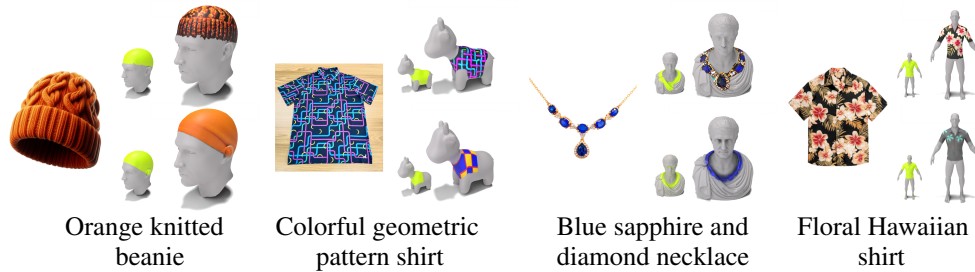

| Orange knitted beanie | Colorful geometric pattern shirt | Blue sapphire and diamond necklace | Floral Hawaiian shirt |

Figure 7: **Qualitative Comparison.** We compare our method (top) to 3D Paintbrush Decatur et al. (2024) (bottom), an approach for text-driven local texture editing, where we extract a detailed text caption from the guidance image via BLIP-2 Li et al. (2023a). Since our method is conditioned on images, we are able to synthesize the desired texture edit with higher precision, especially in cases where the texture edit is difficult to describe with text.

texture renders additionally allows us to update our localization as well. This is important as we want to leverage information about the structure of the reference image to help determine our localization.

Through our masked cross-attention loss, we are able to obtain localizations that reflect the precise structural details of the reference images (see Fig. 5). However, as discussed in Sec. 4.3, when poorly initialized, the localizations learned from the image guidance loss alone can often get stuck in local minima. Thus image guidance alone is not sufficient to set the global location of the texture. To remedy this, we also employ an explicit localization loss driven by text that follows the format of 3D Highlighter Decatur et al. (2023).

### 3.3 TEXT GUIDED LOCALIZATION LOSS

Our localization loss (top of Fig. 4) takes as input the rendered probability predictions and a text prompt that describes the edit broadly (as opposed to describing the details of the reference image). This loss provides coarse guidance to our localization predictions and enables globally coherent placement of our localizations on the mesh. Specifically, we give it the object type of the mesh and the object type contained in the reference image; in the example shown in Fig. 4, the prompt would be "a cow with sunglasses." This generic description allows for supervision that gives a coarse signal for localization, but does not give overly specific guidance that could conflict with the details embodied in our guiding reference image $I$. Using this generic text prompt, we perform an SDS loss with a pre-trained text-to-image model to obtain a gradient for our rendered probability image. This gradient can then be backpropagated through our differentiable renderer to our localization network.

While the inclusion of the text-driven localization loss improves the stability of our learned localizations (see supplementary material), if we simultaneously optimize with our text and image driven losses, we can still fall into local minima due to competition between these two losses (see Fig. 6 "w/o warm up"). To address this, for the first 1000 iterations of our optimization, we only use the localization loss. This localization "warm up" period allows us to obtain a coarse initial guess of our localization before adding in our image-guided loss and greatly improves the robustness of the localizations obtained with our method.

### 3.4 LOCALIZATION MODULATED IMAGE GUIDANCE

**Image Guided Score Distillation with IP-Adapter.** Key to our method is the ability to supervise our local texture edits with image guidance. While standard SDS Poole et al. (2023) approaches typically use text-to-image models, we can apply this loss to joint text and image guided models such as IP-Adapter Ye et al. (2023) as well. The loss still resembles standard SDS, but now given a pretrained IP adapter model $\phi$ we additionally condition on the reference image $I$ as well as the standard conditions of the text prompt $y$, timestep $t$, and rendered image $x$. We obtain the gradient:

$$\nabla_x \mathcal{L}_{\text{SDS}_{\text{IP}}}(\phi, x, y, I) = w(t)(\epsilon_\phi(z_t, t, y, I) - \epsilon) \tag{1}$$

where timestep $t \sim \mathcal{U}(\{1, \ldots, T\})$ is sampled uniformly, noise $\epsilon \sim \mathcal{N}(\mathbf{0}, \mathbf{I})$ is Gaussian, and the noisy image $z_t$ is obtained by applying a timestep-dependent scaling of $\epsilon$ to the image $x$. The weight $w(t)$ is a timestep-dependent weighting function and $\epsilon_\phi(z_t, t, y, I)$ is the noise predicted by the diffusion model.

Figure 8: **Diverse local texture edits of the same shape.** We apply a variety of reference images to edit the same human head mesh. The images contain props of different kinds, shapes, and intricate textures. 3D PixBrush faithfully captures the structure and style of the reference object, and positions it on the correct location on the mesh.

**IP Adapter Joint Text-Image Cross-attention.** Standard text-to-image diffusion models typically condition on text by computing the cross-attention (CA) between the text prompt and UNet features. IP-Adapter enables image guidance by computing an additional decoupled cross-attention between the reference image and the UNet features. To combine the text and image conditions, IP-adapter sums the cross-attention features from text and image. A weighting parameter for the image features controls the influence of the reference image on the output image. For cross-attention features for text $CA_t$ and images $CA_I$ and image weight $w$, the equation for combining the cross-attention features is as follows:

$$CA \text{ Features} = CA_t + wCA_I. \tag{2}$$

While the cross-attention maps for text tokens are well known to have clear correspondence between image regions and specific text tokens Hertz et al. (2022), the image tokens generated by the IP-Adapter do not share this localization property. Instead, the image tokens capture global information about the overall image without containing spatially distinct representations of different regions. This disparity between a localized CA and a global one introduces a non-trivial challenge in linking the two decoupled cross-attention maps, which is essential for our local texture supervision. Since our localizations are based initially on text (due to our localization loss warm up), when we perform this IP-Adapter-based score distillation, we get textures that capture the detail of the guiding image, but are mismatched with our localization (see Fig. 6 "w/o CA mask"). To address this, we propose incorporating our current localization prediction into the cross-attention conditioning process.

**Localization Modulation via Cross-attention Masking.** To modulate our image guidance with our current localization prediction, we extract a 2D binary mask from our localization render at the current view by thresholding the predicted probabilities. We then apply this mask to the cross-attention features at each layer in the UNet to ensure that our image guidance condition is only applied in regions of our image corresponding to our localization. Specifically, we replace Eq. (2) with the following:

$$\text{Masked CA Features} = CA_t + wCA_I M_l \tag{3}$$

where $M_l$ is our 2D localization mask $M$ downsampled to the resolution of the features in the given UNet layer $l$. Similarly, our new localization-masked IP-Adapter SDS gradient is computed using:

$$\nabla_x \mathcal{L}_{\text{SDS}_{\text{MaskIP}}}(\phi, x, y, I, M) = w(t)(\epsilon_\phi(z_t, t, y, I, M) - \epsilon) \tag{4}$$

where $M$ is again the 2D localization mask. This whole process is illustrated in Fig. 4 within Localization Modulated Image Guidance.

This masked cross-attention score distillation gives much more localized image supervision; since the image-guided supervision is aware of the localization prediction, we obtain textures which are better aligned to our localizations (Fig. 6). In addition to improved textures, this approach also facilitates the incorporation of the precise structure of our reference image into our localization predictions (see Figs. 5 and 10). In summary, standard IP-Adapter image guidance is global, but we are able to tailor it to our local texture edits by modulating the image cross-attention features with our predictions of the localization region.

## 4 EXPERIMENTS

We demonstrate the effectiveness of 3D PixBrush through a variety of experiments. Sec. 4.1 illustrates several key properties of 3D PixBrush. In Sec. 4.2 we compare 3D PixBrush, qualitatively and quantitatively, with a recent local mesh editing baseline Decatur et al. (2024). In Sec. 4.3 we conduct ablations demonstrating the importance and influence of the method's components.

## 4.1 PROPERTIES OF 3D PIXBRUSH

**Generality.** 3D PixBrush handles a variety of meshes and reference images as input, as shown in Figs. 1, 3 and 8 (*e.g.* dressing an animal in a neon shirt and placing a diamond necklace on a Napoleon bust). Our method is not limited to guidance from a specific class of reference objects. Within these classes, our reference objects span diverse shapes and styles such as our garments of varying pattern, length, and formality and the many facial accessories we show. Our target meshes may have smooth geometry, like the cow in Fig. 1 and the hand in Fig. 3, or include intricate geometric details, like the human head in Fig. 8 and carved bust in Fig. 3.

In all these cases, our method successfully synthesizes local textures, even when the shape and image combination is out-of-distribution. Although animals do not usually wear sunglasses or a mask, and there is no geometry of eyes or mouth for the animal in Fig. 1, 3D PixBrush places the sunglasses and the mask in a plausible location.

**Specificity.** 3D PixBrush demonstrates high specificity in several aspects: the location of the edit, its shape, and its texture (see Figs. 3 and 8). First, we observe that the object in the reference image is placed in the correct position region on the shape. For example, watches and bracelets are located on the wrist, scarves and necklaces around the neck, and the blue headphones cover the ear region.

Our method is also faithful to the *shape and texture* of reference image. In Fig. 3, the denim jacket has long sleeves, the flower T-shirt has short ones, and the construction vest has none. Accordingly, the local texture edit on the human body covers, respectively, the entire arms, part of the arms, and only the torso, adhering the to reference image with high granularity. Additionally, our texture edits capture the colors in the reference image as well as intricate patterns such as the floral pattern. Our method even distinguishes between objects with a similar shape and location, but differing colors, as seen in Fig. 3. For example, both watches have a circular shape and are placed on the wrist, yet our method successfully synthesizes the green or white interior color. Similar phenomena can be observed for the eye accessories in Fig. 8. We attribute the specificity of 3D PixBrush to our localization-modulated image guidance.

**Robustness.** The optimization process of our local texture edit starts by initializing a coarse localization utilizing only text and continues with further refinement of the region and generation of the conforming texture using the additional image guidance. The text does not contain information about the shape of the accessory depicted in the image, and the initial localized region does not match the target edit (see Fig. 10 in the appendix). Nonetheless, our method overcomes this discrepancy and successfully updates the region to align with the desired asset style in the image.

Surprisingly, we have found that we can even skip the warm-up phase and initialize the local region with a *fine-grained* localization based on a shape different from the object in the reference image, and the final result still converges to adhere to the image correctly. In Fig. 10, the localization starts from distinct heart-shaped sunglasses obtained from the prior shape editing work 3D Paintbrush Decatur et al. (2024). Nonetheless, the region is changed to a circular, an oval, or a star shape, matching the corresponding image. This robustness is enabled by our novel localization-modulated image guidance module, which updates the local edit region and aligns it with the image.

## 4.2 EVALUATION

**Qualitative comparison.** We compare 3D PixBrush with a recent method, 3D Paintbrush Decatur et al. (2024). 3D Paintbrush uses a text prompt to generate a local texture for an input mesh. For a fair comparison, we use a state-of-the-art image captioning model BLIP-2 Li et al. (2023a) to obtain detailed text descriptions of the guiding images and use them as input to 3D Paintbrush.

Fig. 7 shows the results of our method and 3D Paintbrush for the guidance image and the corresponding BLIP-2 text prompt, respectively. 3D Paintbrush localizes the edit and colors it reasonably according to the text prompt. However, it does not adequately produce the desired texture. For the "Orange knitted beanie" example, 3D Paintbrush's edit is indeed orange, but although the knitted pattern is mentioned in the text, it is missing in the result. In contrast, 3D PixBrush faithfully generates both the color and texture of the object in the reference image.

Moreover, some textures are very hard to describe precisely only with text, such as the "Colorful geometric pattern shirt." In this case, 3D Paintbrush does create an edit that matches the text descrip-

tion but is completely unrelated to the *particular* colorful geometric pattern depicted in the image. 3D PixBrush, on the other hand, accurately matches the color and structure of the desired pattern, demonstrating its advantage over 3D Paintbrush. A similar phenomenon is also observed for the shirt with the unique flower pattern which is successfully captured by our method.

**Quantitative evaluation with CLIP R-Precision.**
We quantitatively compare our method to 3D Paintbrush Decatur et al. (2024) using CLIP R-precision. This uses CLIP Radford et al. (2021) to measure the alignment between the renders of our generated local edit and the reference image used to generate them. The reported numbers in Tab. 1 correspond to the percentage of generated results that are able to successfully retrieve the image that was used to guide their generation. 3D PixBrush consistently outperforms 3D Paintbrush on all CLIP models.

Table 1: Quantitative evaluation. We compare our local edits to 3D Paintbrush Decatur et al. (2024) (3DPB) and report CLIP R-Precision.

| Method | CLIP R-Precision ↑ | | |
|---|---|---|---|
| | CLIP B/32 | CLIP B/16 | CLIP L/14 |
| 3DPB | 28.57 | 33.33 | 38.10 |
| **Ours** | **52.38** | **47.62** | **71.43** |

### 4.3 ABLATION STUDY

We demonstrate the importance of the design choices in our method with two ablation experiments presented in Fig. 6. First, we omit the localization masking of the cross-attention between the U-Net layers and the reference image in our Localization Modulated Image Guidance component (depicted in Fig. 4). Without the masking, we can still roughly obtain the texture of the reference object, but the region and the location of the edit are incorrect. The cross-attention masking in our Localization Modulated Image Guidance focuses the interaction between the rendered mesh and the reference accessory in the Denoising U-Net model to the edit region only, reducing the influence of the other mesh parts that compromises the edit result.

Additionally, we conduct an ablation turning off the text-only localization warm-up and optimizing the localization and texture from scratch using both the text and image guidance. In this case, the localization completely fails, as seen in Fig. 6. This experiment reveals an interesting insight - the influence of the text and image on the local texture is mainly decoupled. Text has a strong spatial influence on the texture region and is the dominant factor in determining its global location on the mesh. In contrast, the image guidance allows us to capture the fine-grained details of the reference image texture and, through our LMIG, is able to refine the localization to match the precise structure of the reference image, as witnessed in Figs. 5 and 10. However, on its own, the image guidance struggles to set the global location of the edit.

**Limitations.** In Fig. 11 in the appendix, we show several limitations of our method. In cases where our reference image contains writing, our method captures the essence of the reference image, but can struggle to reconstruct the text exactly. For example, a reference image depicting a red flip flop contains the brand name "adidas." However, our method synthesizes the text as "adadds." In cases where the local edit can semantically apply to other regions on the mesh, these other regions are sometimes also localized and textured by our method. While the reference image shows makeup applied only to the regions around the eyes, given the semantic connection between makeup and lipstick, our method applies this texture to the lips of the head mesh in addition to the eyes. Our method also occasionally suffers from the 'Janus' effect wherein 3D generative methods generate multiple front-facing views (*i.e.* adding sunglasses on the front and back of the head).

## 5 CONCLUSION

We presented a technique for synthesizing local texture edits on meshes using images as guidance. Our method is capable of automatically positioning the asset onto the input mesh in a *globally coherent manner*. Moreover, our predicted localizations are sufficiently fine-grained such that they reflect the various geometric structures depicted in the reference image. Our predicted textures adhere to the object in the reference image. Key to achieving these results is our proposed localization modulated image guidance. In the future, we are interested in exploring additional applications of our proposed localization modulated image guidance to different domains, such as images, NeRFs Mildenhall et al. (2020), and videos.

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

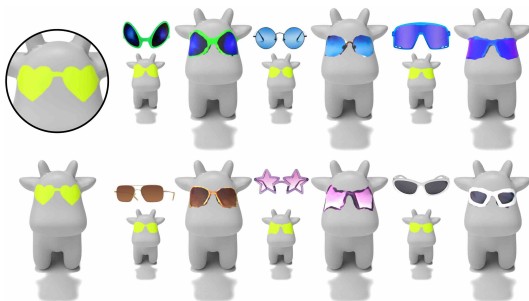

Figure 9: **Additional Results.** We show results of our method on additional diverse meshes. Our method automatically predicts a fine-grained segmentation mask that accurately reflects a plausible placement of the reference image on the mesh (*e.g.* the overalls cover the legs and go over the shoulders and the scarf is draped around the neck). These results contextualize the reference image and synthesize a plausible variant of it that conforms to the input geometry and to the local predicted segmentation mask (see for example the colorful poncho on the left). Our method also captures the texture detail in the overall jeans (see for example the green line present in the result).

## A  ROBUSTNESS

Fig. 10 demonstrates the robustness of our method to the localization initialization. The local region can start from a given specific shape different than that of the reference image object and still converge to a region corresponding to the asset in the guidance image.

Figure 10: **Robustness.** 3D PixBrush is robust to the initial localization region. Here, the localization starts out as heart-shaped sunglasses (left), which is different than the shape of the sunglasses in the guidance image examples shown. Still, our method overcomes this discrepancy and corrects it to match the reference image.

## B  LIMITATIONS

In Fig. 11, we show the limitations of our method. 3D PixBrush struggles to reproduce text depicted in the guidance image. Additionally, there may be semantic coupling in the edited result, where regions of the mesh closely related semantically to the asset in the image are colored as well.

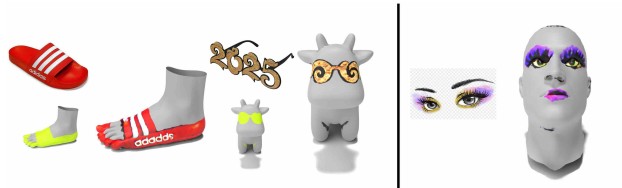

Figure 11: **Limitations.** Our method captures the style of the reference image but can struggle to exactly reproduce text (left). In cases where the guiding image carries strong semantic connotations, components that are closely semantically related to the guiding image, but not included in the image can be included in the localization (right). For example, in this image that contains makeup, lipstick is also included in our edit.

# C  ADDITIONAL QUANTITATIVE EXPERIMENTS

We show additional quantitative comparisons between our method and 3D Paintbrush using a perceptual study, additional CLIP metrics, and the LPIPS metric.

**Perceptual Study.** We conduct a perceptual study in which 34 users rate the effectiveness of both our method and 3D Paintbrush on 10 different mesh and reference image combinations. The users are asked to evaluate the quality of both the edit texture (the actual colors and patterns) and edit structure (the shape and location on the object). 3D PixBrush (ours) scores higher than 3D Paintbrush, producing more accurate textures and structures (see Tab. 2).

Table 2: Perceptual Study. We conduct a perceptual study where users evaluate the texture and structure of our local edits compared to 3D Paintbrush Decatur et al. (2024).

| Method | Texture Avg. Score ↑ | Structure Avg. Score ↑ |
|--------|--------|--------|
| 3DPB | 1.96 | 2.69 |
| **Ours** | **4.11** | **4.17** |

**Text-based CLIP R-Precision.** In addition to the image-based CLIP R-Precision that we report in the main paper, we run a text-based CLIP R-Precision evaluation as well (see Tab. 3). Instead of using the renders of the generated results to retrieve the reference images, we use the renders to retrieve the BLIP 2-generated captions. We find that our method still outperforms 3D Paintbrush Decatur et al. (2024) despite the fact that our method is only conditioned on the reference images and does not take the captions as input, while 3D Paintbrush is *directly* conditioned on these exact BLIP 2 captions. We hypothesize that this is due to the fact that text-driven approaches like 3D Paintbrush don't capture all sentiments contained in the text prompt equally well and tend to neglect certain components. In contrast, our image-driven approach better captures the global style of the reference image. Since the BLIP 2 captions (which are used as text prompts) are derived from the reference images, 3D PixBrush captures more components of the text prompts.

Table 3: Text-based CLIP R-Precision. We compare our local edits to 3D Paintbrush Decatur et al. (2024) (3DPB) and report a text-based CLIP R-Precision metric using BLIP 2 Li et al. (2023a) captions of the reference images. Our method outperforms 3D Paintbrush on all CLIP models, despite the fact that our method is guided by the reference images, whereas 3D Paintbrush directly uses these BLIP 2 captions as text prompts to guide their optimization.

| | Text-based CLIP R-Precision ↑ | | |
|--------|--------|--------|--------|
| Method | CLIP B/32 | CLIP B/16 | CLIP L/14 |
| 3DPB | 76.19 | 76.19 | 80.95 |
| **Ours** | **90.48** | **80.95** | **85.71** |

**CLIP Similarity Metric.** In addition to the R-Precision metrics, we compare our method to 3D Paintbrush Decatur et al. (2024) by directly reporting average CLIP similarity scores in Tab. 4. We take the renders of the generated results for each image and encode them into CLIP's latent space. We also encode the reference images. We then compute the cosine similarity in CLIP latent space between the generated renders and their corresponding reference images and report the average similarity over all examples. 3D PixBrush achieves higher similarity scores than 3D Paintbrush on all CLIP models.

**LPIPS Metric.** In Tab. 5, we evaluate our method as compared to 3D Paintbrush using the LPIPS Zhang et al. (2018) metric. This metric uses deep features of visual models to measure similarity between images. Here, lower scores correspond to higher similarity. We find that 3D PixBrush achieves lower LPIPS scores and thus higher similarity with the reference images than 3D Paintbrush using deep features from both AlexNet Krizhevsky et al. (2012) and VGG Simonyan (2014).

Table 4: CLIP Similarity Scores. We compare our local edits to 3D Paintbrush Decatur et al. (2024) (3DPB) and report the average cosine similarity in CLIP embedding space between renders of the generated results and their corresponding reference images. Our method receives higher average similarity scores than 3D Paintbrush on all CLIP models.

| | CLIP Similarity Scores ↑ | | |
|---|---|---|---|
| Method | CLIP B/32 | CLIP B/16 | CLIP L/14 |
| 3DPB | 0.61194 | 0.62860 | 0.57538 |
| **Ours** | **0.65325** | **0.66936** | **0.61043** |

Table 5: LPIPS Quantitative Evalution. We compare our local edits to 3D Paintbrush Decatur et al. (2024) (3DPB) and report the LPIPS Zhang et al. (2018) perceptual metric between the generated results and their corresponding reference images. Our method outperforms 3D Paintbrush on both AlexNet and VGG.

| Method | LPIPS AlexNet ↓ | LPIPS VGG ↓ |
|---|---|---|
| 3DPB | 0.63988 | 0.52314 |
| **Ours** | **0.63575** | **0.51803** |

## D IMPLEMENTATION DETAILS AND ABLATIONS

**Supervision and Optimization Details.** For our localization loss, we employ DeepFloyd IF AI (2023) and run it for 1000 optimization steps. We then continue to update the localization region as well as optimize the local texture edit according to the reference image using both DeepFloyd and IP-Adapter Plus Ye et al. (2023) for an additional 10000 steps. During these additional 10000 iterations, we still use the localization loss as well. 3D PixBrush is implemented in PyTorch Paszke et al. (2017). We run our experiments using an NVIDIA A40 GPU. For the results shown in the paper, we run our local texture optimizations for 4 hours, however, we obtain reasonable results after roughly 1 hour (see Fig. 12).

**View Selection Details.** In order to supervise our optimization, we render 2D images of our 3D mesh. The camera parameters for these renders are randomly sampled each iteration of the optimization. By default, camera elevation values are sampled between 0 and 60 degrees, azimuth values are sampled from the range $0 - 360$ degrees, and radius values are sampled from the range $1 - 1.5$. For a small subset of meshes that are predominantly viewed from above, we expand the elevation range to $0 - 150$.

**Neural Network Architecture Additional Details.** As specified in Sec. 3.1, we use MLPs consisting of a positional encoding layer followed by 6 fully connected layers for both the neural localization and neural texture networks. The positional encoding layer in the neural localization network uses a $\sigma$ value of 6 while the positional encoding in the neural texture network uses a $\sigma$ value of 12 to allow for the higher frequency textures we output. The middle layers of the MLPs have width 256

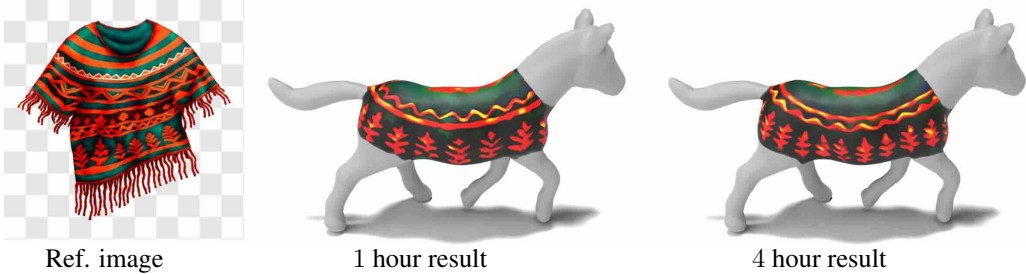

| Ref. image | 1 hour result | 4 hour result |

Figure 12: **Optimization Run Time.** We show the results of our approach after running the optimization for both 1 hour (middle) and a full 4 hours (right). In many cases, our approach achieves satisfactory results after only 1 hour that are comparable to the full 4 hour results.

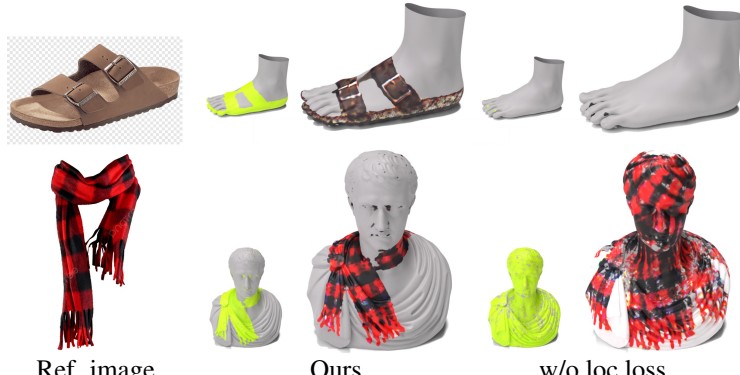

Ref. image      Ours      w/o loc loss

Figure 13: **Localization Loss Ablation**. 3D PixBrush (Ours) produces accurate localizations and detailed textures, each of which matches the structures and styles respectively of the reference images. Removing the localization loss (w/o loc loss) often results in trivial localizations that segment all of the mesh and global textures that ignore the geometry.

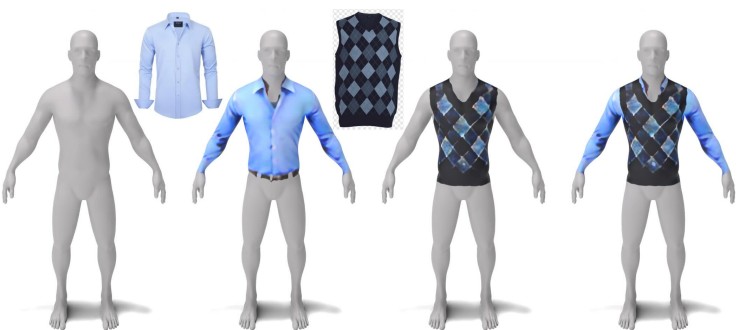

Figure 14: **Composition of Local Texture Edits.** 3D PixBrush produces precise localizations to accompany the predicted local textures. A key strength of predicting an *explicit fine-grained segmentation mask* is that it enables an additional level of control for graphics applications. For example, it is possible to trivially composite multiple local textures on the same mesh. Another application of our predicted localization masks is shown in Fig. 15.

and the output layers produce 1 value (probability) for the localization network and 3 values (RGB) for the texture network.

**Localization Loss Ablation.** In Fig. 6 in the main paper, we show that removing the localization loss warm up period degrades performance. Here, we further investigate the importance of the localization loss by removing it entirely. In Fig. 13, we see that removing the localization loss entirely still results in degraded performance: either all or none of the region is localized, and textures are applied globally instead of locally.

# E ADDITIONAL RESULTS

**Composition of Local Texture Edits.** In Fig. 14, we show composited local textures using 3D PixBrush. Since our method produces precise localizations that accompany our local textures, we can easily composite one texture on top of another. We first locally texture the person to add the dress shirt (middle left). Then we independently locally texture the person to add the sweater vest (middle right). Using the localizations for each respective local texture, we composite the sweater texture on top of the dress shirt texture and add both to the person in a coherent way (far right).

**Edits Over Existing Textures.** In Fig. 15, we use our method to generate local textures on a bare mesh and overlay them on an existing global texture (left) for that mesh. The precise localization masks predicted by our method enable us to trivially composite our local textures onto the mesh such

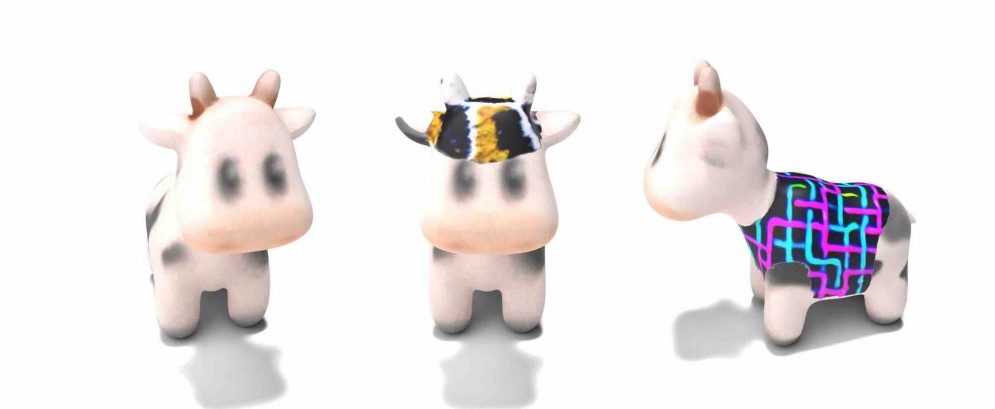

Figure 15: **Compositing Over Existing Textures.** Starting with an existing global texture (left), we can use our precise localization masks to trivially composite our local textures onto the mesh. Our predicted localization masks are precise enough that they enable effectively matting the predicted texture while leaving the original cow texture unchanged. This application is trivially enabled due to our deliberate choice in predicting a fine-grained segmentation mask.

that we only change regions corresponding to the local texture and preserve the rest of the existing global texture on the mesh (middle, right).

**Same Reference Image Applied to Diverse Meshes.** In Fig. 16, we apply local textures driven by the same reference image (left) to multiple diverse meshes. In each case, the style of the reference shirt is effectively applied to the mesh. Although these meshes have very different shapes, 3D PixBrush takes the context of the target mesh into account and properly structures the local texture in a way that makes sense on the given mesh.

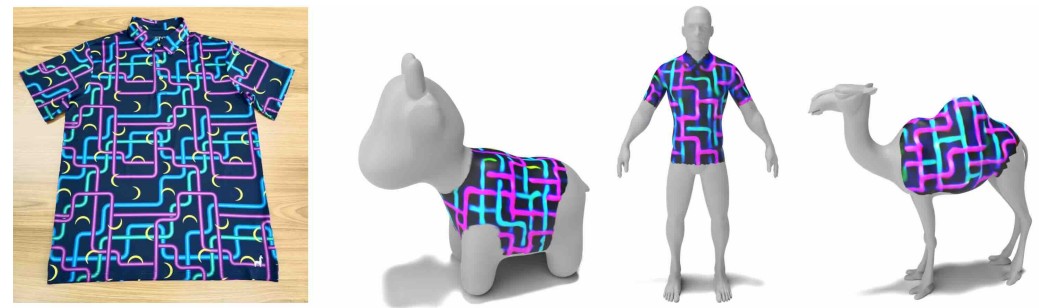

Figure 16: **Same Reference Image Applied to Diverse Meshes**. 3D PixBrush can produce local textures on a diverse set of meshes using the same reference image. Our method is capable of accurately contextualizing where the reference image ostensibly belongs on a variety of different input shapes.

**Local Textures in Different Contexts.** In Fig. 17, we apply a local texture driven by the same reference image (top left) in multiple different contexts. In each case, the style of the reference shirt is effectively applied to the mesh. Although these meshes have very different shapes, 3D PixBrush takes the context of the target mesh into account and properly structures the local texture in a way that makes sense on the given mesh geometry. For example, the first two meshes have a larger upper torso than the third mesh. On these two meshes, our method generates a larger shirt that is tight-fitting in this upper torso region, but is looser and, as a result, more wrinkled in the lower region. In contrast, the third mesh, which is more slender is given a smaller, tighter-fitting shirt that displays fewer wrinkles in the lower section.

**Local Geometric Deformation.** In Fig. 18, we combine our local image-driven texture edits with an off-the-shelf deformation framework Dinh et al. (2025) to achieve deformations in just the region

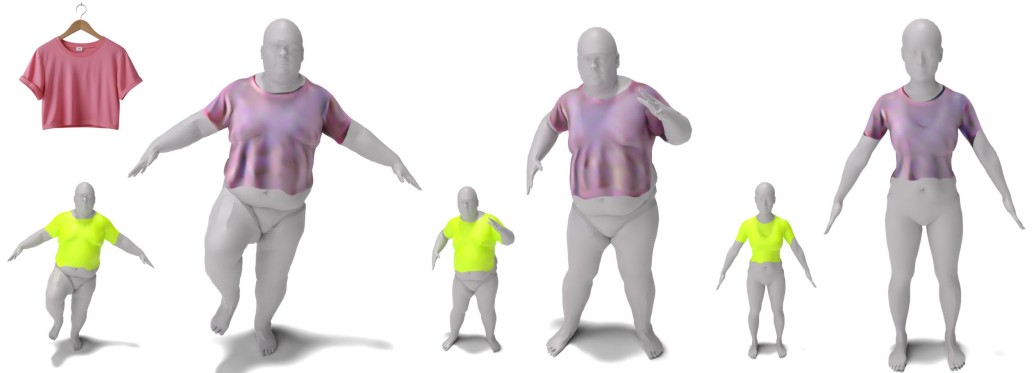

Figure 17: **Local Texture in Different Contexts.** Using a reference image of a pink crop top (top left), 3D PixBrush produces consistent localizations and local textures that capture the details of the shirt in the reference image and adapt it to fit the context across different poses of the same mesh or even different geometries altogether. This result highlights that this task requires a solution beyond a simple cut-and-paste of the reference image. We see that our method synthesizes wrinkles, texture details, and material changes to conform to different meshes.

of the local edit. Once we obtain our local edit, we use our learned localization mask to direct the deformation to the precise area of the local edit to obtain a local edit with both texture and geometric components.

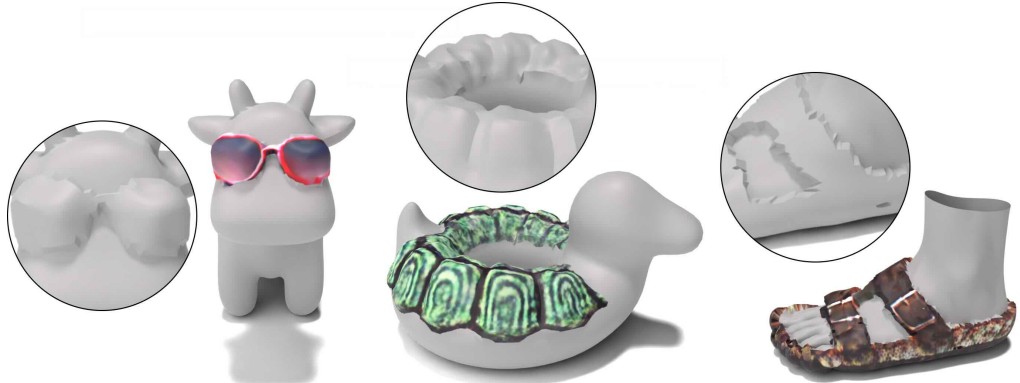

Figure 18: **Local Deformation Application.** Once we have obtained our local image-driven texture edits, we can apply an off-the-shelf mesh deformation framework Dinh et al. (2025) only in the region of our localization mask to obtain a local edit with both texture and geometric components.

**Result on Genus One Mesh.** In Fig. 19, we use the reference image of a turtle shell to learn a local texture on this rubber duck using our method. Even though the mesh is genus one and turtle shells are typically seen on flat, continuous surfaces, our method is able to adapt the local texture to the unique shape of the mesh. 3D PixBrush produces a local texture that wraps around the hole on the mesh's back while still displaying a pattern that matches the shell in the reference image.

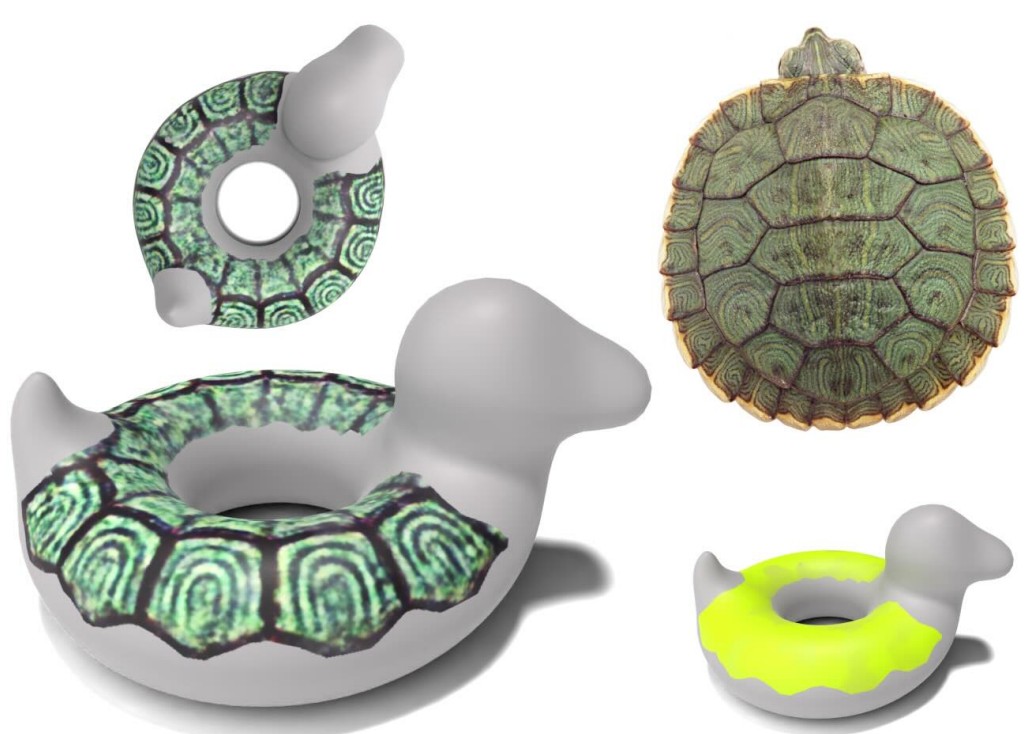

Figure 19: **Result on Genus One Mesh.** We show the result of our approach using the turtle shell reference image (top right) to produce a local texture on this genus one rubber duck mesh. Even though turtle shells are typically seen on flat / genus zero surfaces, 3D PixBrush is able to produce a plausible shell texture that conforms to the unique shape of this mesh by properly wrapping around the hole on its back.

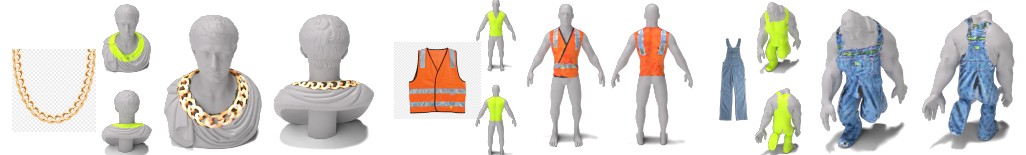

Figure 20: **Multi-View Results.** We show results of our method from multiple views. 3D PixBrush generates 3D consistent local textures that look plausible from arbitrary angles.