# OpenReview forum: "3D PixBrush: Image-Guided Local Texture Synthesis"
_ICLR.cc/2026/Conference — Submitted to ICLR 2026_

### Official Review · Reviewer_QuJ6 · 2025-10-26

**Soundness:** 3
**Presentation:** 3
**Contribution:** 2
**Rating:** 6
**Confidence:** 3

**Summary:**

The paper introduces 3D PixBrush, a method for locally editing 3D meshes by synthesizing textures guided by a reference image. It leverages a learned localization mask to accurately position the texture on the mesh and then synthesizes the texture to adhere to the style of the reference image. The method uses a modified score distillation sampling technique with both predicted localization and the reference image to provide guidance. It requires no manual user input (scribbles or bounding boxes) to achieve accurate localizations. The paper shows qualitative and quantitative results on a variety of meshes.

**Strengths:**

The strengths include:
-The approach of using a reference image to both localise and texture a specific region of a 3D mesh without manual user input is a new and interesting idea in texture editing.

-The paper generally explains the methodology well, particularly the Localization Modulated Image Guidance (LMIG) component.

-The figures demonstrate the method's ability to transfer textures from reference images to 3D meshes in a semantically meaningful way. Some of the examples (e.g., the diverse textures on the same human head) are visually pleasing

-The paper includes quantitative evaluations using CLIP R-precision, showing improvements over the CVPR 2024 baseline method.

**Weaknesses:**

-The description of some components lacks sufficient technical detail. For instance, it would be good to specify the architecture of the Neural texture network in section 3.1.

-ablation studies are present, they do not analyze all of the hyperparameters that exist.

-The evaluation metric CLIP R-Precision may not fully capture the perceptual quality of the results.

**Questions:**

-Please provide more technical details about the Neural Texture network and the structure for the text guided 3D Generation component.

-The method predicts localization masks - how are these handled to avoid sharp transitions and artifacts when applying the synthesized textures?

-Please clarify what specific novel contributions does the paper make on top of other global style transfer methods?

---

> ### Author Response · Authors · 2025-11-21
>
> We thank the reviewer for their insightful and thorough feedback. We address their comments below.
>
> **W1: Request for additional technical details**
>
> We provide additional details of our neural architecture in the revised PDF (L846).
>
> **W2: Current ablation studies do not analyze all hyperparameters that exist**
>
> We include ablations on both our localization cross attention masking (LMIG) and the warm up stage (see Fig. 6 in the main paper) as well as an ablation on the localization loss (Fig. 13 in the supplemental, L876). We chose to formally ablate design choices and hyperparameters that constitute key contributions of our method and/or ones that we observed have the most impact on result quality. If there are specific additional ablations that would aid the reviewers in evaluating our work, we would be happy to run them.
>
> **W3: Quantitative evaluation of perceptual quality.**
>
> In addition to the CLIP R-Precision metric reported in the main paper, we include a perceptual study in the supplemental material (in original PDF, after references, L761) in which users are asked directly about the perceptual quality of the edited results. In the supplemental we also include evaluation using the LPIPS metric (L805) which gives an additional perspective on perceptual quality using features from an image model. Additionally, we now include an evaluation using VQA score [1] below which is a further measure of perceptual quality. For this metric, we use the same 21 mesh and reference image combinations. The VQA score is computed according to [1] which asks a VQA model the question “Does this figure show \<text\_description\>? Please answer yes or no.” and then outputs the decoder’s predicted probability of “yes.” Our text descriptions are created by using the format “A 3D render of a gray \<object\_name\> with \<BLIP2 caption of reference image\>.” We note that using BLIP2 to caption the reference image often does not perfectly capture all of the subtleties of the image (for example see the BLIP2 generated captions in Fig. 7). However, this only favors the baseline method 3D Paintbrush since their local edits are generated directly from text prompts containing these BLIP2 captions that we evaluate VQA on. Still, our approach scores significantly higher on this VQA metric.
>
> | Method        | VQA Score |
> |---------------|-----------|
> | 3D PaintBrush | 0.48      |
> | **Ours**      | **0.62**  |
>
> **Q1: Additional details on neural texture**
>
> Per W1, we have added additional details on our neural texturing in the revised PDF (see L846).
>
> **Q2: How do we avoid sharp transitions in the localization masks?**
>
> We learn a localization mask over the course of the optimization in a way such that our predicted local textures are a function of our current mask prediction. This ensures that our synthesized textures match up with our predicted localization masks, thus preventing mismatches in the localization masking and avoiding sharp transitions.
>
> **Q3: Contributions relative to global texturing methods**
>
> We focus on the task of _local_ image-guided texturing in that we want to both _identify_ a localization region and _locally texture_ it in a way that conforms to that region while leaving everything outside exactly as it was. While there exist global methods for image-guided texturing, these methods do not provide a way to a) produce an explicit localization and b) force the image-guidance to generate an edit that respects the context of the local region. To address a), we use the concept of a localization neural field guided by an image. For b) we propose our _novel_ LMIG module which enables us to localize the superversion from our image guidance to only our predicted localization region.
>
> **References:**
> [1] Lin, Zhiqiu, et al. "Evaluating text-to-visual generation with image-to-text generation." European Conference on Computer Vision. Cham: Springer Nature Switzerland, 2024.

---

> > ### Comment · Reviewer_QuJ6 · 2025-11-28
> >
> > Thanks for answering all my questions.
> >
> > I appreciate the revisions in the updated paper. My concerns have been addressed.
> >
> > After reading other reviewers' comments, I agree with the argument that the novelty comparing to existing works, e.g., 3D PaintBrush. The answers provided by the authors are not satisfactory.
> >
> > Overall, I will not argue for accepting this paper, but I maintain my score since I still think this work is interesting and the proposed approach is effective.

---

> > > ### Author Response · Authors · 2025-12-03
> > >
> > > We are happy to hear that QuJ6’s concerns have been addressed and that they believe this work is both interesting and effective. Regarding the similarities between our approach and 3D Paintbrush, we reiterate that 3D paintbrush *only supports text-driven* local texturing, while we enable *image-guided* local texturing. Text-driven local texturing does not have to deal with challenges such as: 1) how to obtain a localization mask that adheres to the reference image and 2) how to get the essence of the reference image to be applied only to the localization region and not globally. While 1) can be partially addressed by using image-guided SDS, 2) requires the technical contribution of our LMIG. Using our approach enables us to produce local, image-guided textures that much better adhere to the reference image than what can be achieved using 3D Paintbrush which is guided by text (see Fig. 7).

---

### Official Review · Reviewer_o6on · 2025-10-31

**Soundness:** 2
**Presentation:** 3
**Contribution:** 1
**Rating:** 2
**Confidence:** 4

**Summary:**

The paper proposes a novel framework for image-guided local texture editing on 3D meshes. The method leverages Score Distillation Sampling (SDS) loss to jointly optimize two neural field representations: a localization map and a texture field.
Unlike prior works that either require users to manually define the localization map or rely solely on text prompts for guidance, this approach integrates the IP-Adapter to enable reference-image-based texture generation. Moreover, to ensure that the generated textures remain consistent with the global localization structure while preserving fine-grained local details, the authors introduce a Localization Map–Integrated Guidance (LMIG) mechanism, which masks the attention operation in the IP-Adapter using the predicted localization map.

**Strengths:**

1.It further introduces a Localization Map–Integrated Guidance (LMIG) mechanism, which integrates a mask into the cross-attention module of the IP-Adapter to ensure that the generated texture aligns with the predicted localization map.

2.Extensive experiments demonstrate the effectiveness and superior performance of the proposed approach.

**Weaknesses:**

1.The novelty of this work is limited, as the IP-Adapter used for image conditioning is already widely adopted in the community, and the proposed LMIG merely incorporates a predicted mask into the attention map.

2.The paper employs the IP-Adapter as the image guidance mechanism, extending 3D PaintBrush from text-guided editing to image-guided texture generation. However, since 3D PaintBrush already predicts both the localization map and the local texture, the novelty of this extension is rather limited and not clearly distinguished.

3.In the proposed LMIG mechanism, there is no analysis or justification provided regarding why the masked attention map can improve the consistency between the localization map and the texture. The initial localization map is obtained through a text-guided loss, which may not be fully aligned with the reference image content. Despite this potential mismatch, the localization map is directly used to modulate the attention map, while the final output simultaneously influences the localization map itself. This design introduces a potential circular dependency and raises questions about the stability and interpretability of the proposed mechanism.

**Questions:**

1.The CMD[1] framework performs multiview 3D editing, jointly modifying both the mesh geometry and texture, where local texture editing is treated as a sub-process. Could the authors elaborate on the advantages of their method compared with CMD? It would also be informative to evaluate the proposed approach on some of the cases used in the CMD paper.

2.I am curious about how the LMIG mechanism behaves when the text prompt does not correspond well to the reference image, and how the initial localization mask influences the final results under such conditions.

3.Since the image-guided loss also updates the localization map, how would an inaccurate mask applied in the attention operator affect the optimization of the localization map?

\[1\] [https://dl.acm.org/doi/10.1145/3721238.3730722](https://dl.acm.org/doi/10.1145/3721238.3730722)

---

> ### Author Response · Authors · 2025-11-21
> **Official Comment by Authors (1/2)**
>
> We thank the reviewer for their insightful and thorough feedback. We address their comments below.
>
> **Response to weaknesses**
>
> **W1: “proposed LMIG merely incorporates a predicted mask into the attention map”**
>
> We argue that using our LMIG predicted attention masks are a unique and interesting technical contribution. Using a _continually updated learned mask_ has not been done before, and doing so allows us to contextualize our image guidance to be contained within the areas we want to locally texture. If we instead use a naive approach of simply masking the texture with the localization after we generate it, this results in a mismatch between the localization and the texture (see Fig. 2 and W3 response below for more details).
>
> **W2: Distinction from 3D Paintbrush**
>
> 3D paintbrush _only supports text-driven_ local texturing, while we enable _image-guided_ local texturing. Text-driven local texturing does not have to deal with challenges such as: 1) how to obtain a localization mask that adheres to the reference image and 2) how to get the essence of the reference image to be applied only to the localization region and not globally. While 1) can be partially addressed by using image-guided SDS, 2) requires the technical contribution of our LMIG.
>
> **W3:**
> > “In the proposed LMIG mechanism, there is no analysis or justification provided regarding why the masked attention map can improve the consistency between the localization map and the texture.”
>
> Fig. 2, right side, middle column shows that without masked attention the texture is not localized (the sunglasses cover the entire side of the cow and are not on the eyes). Masking this global texture with the localization mask that covers the eye region will produce a meaningless result since the sunglasses texture is not contained to the eye region. If we instead mask the attention, we are able to produce textures that conform to our localization region, thus ensuring that they are consistent when we later explicitly mask with our localization.
>
> > “The initial localization map is obtained through a text-guided loss, which may not be fully aligned with the reference image content.”
>
> While the initial localization is obtained using text guidance, it is refined using the image guidance so that we obtain a localization mask that is influenced by the reference image (see in Fig. 5 how the vest localization for our method conforms to the exact shape of the reference vest, while the “text only” case generates a generic vest localization that does not match the reference vest style). Thus, our localization map _is_ aligned with our reference image content. In this sense, we use our image-guided localization map to influence our attention in the global style prediction and thus they match up.
>
> > “the localization map is directly used to modulate the attention map, while the final output simultaneously influences the localization map itself. This design introduces a potential circular dependency and raises questions about the stability and interpretability of the proposed mechanism.”
>
> There is interplay between the simultaneous optimizations of the localization and local texture. The localization is used to mask both the globally predicted texture as well as the attention in our LMIG during the texture optimization. Additionally, gradients from our texture loss are back propagated to the localization during its optimization. We argue that this simultaneous optimization is a key strength as it helps to ensure alignment between the predicted localization and local texture.

---

> > ### Author Response · Authors · 2025-11-21
> > **Official Comment by Authors (2/2)**
> >
> > **Response to questions**
> >
> > **Q1: Advantages compared to CMD [Li et al., SIGGRAPH 2025]**
> >
> > A key difference is that our method learns an explicit localization mask, whereas CMD does not. Since our method explicitly predicts a localization mask, we can guarantee that anything outside of that mask will not be changed. CMD does not have this guarantee since it relies on a target edit image (a locally edited rendered image of the mesh) as input to encode the local edit. Furthermore, our explicit localization masks allow us to trivially composite multiple local textures together on the same mesh without worrying about the background components overwriting other local textures (see Fig. 14 in the supplemental, L892). Since CMD does not have explicit edit localizations, in order to create multiple edits on the same mesh, those edits would need to be added sequentially which does not allow for instant mixing and matching of edits like ours does.
> >
> > **Q2: What happens when the text does not conform to the reference image?**
> >
> > In all our results, the text prompts we use only include a generic name for the object in the reference image. For example in Fig. 1, the round pink sunglasses with gradient lenses and the neon shirt with a detailed pattern on it are referred to simply as “sunglasses” and “shirt” respectively. Thus, our text prompts are already not a perfect match for the reference image. Furthermore, the text is primarily used to obtain an initial guess for the localization mask and our method supports refining masks to conform to the reference image. Fig. 10 in the supplemental material (L718 in the PDF) shows our method starting from an initial user-provided mask (heart shaped sunglasses) that does not match any of the reference images. We use our image guidance to update the mask to better adhere to the object shape in the reference images.
> >
> > **Q3: How would an inaccurate mask affect the LMIG?**
> >
> > Our method is robust to initializing our optimization with a mask that does not match the target in our reference image. Fig. 10 shows our method starting with an inaccurate mask (which is used in the LMIG) and still obtaining localization masks that better adhere to the reference image.

---

### Official Review · Reviewer_4fRa · 2025-11-01

**Soundness:** 3
**Presentation:** 3
**Contribution:** 3
**Rating:** 6
**Confidence:** 4

**Summary:**

This paper introduces 3D PixBrush, a novel method for image-guided local texture synthesis without user-specified mask, specifically on 3D meshes. The approach leverages a reference image and a text prompt to predict both a localization mask and a corresponding texture, enabling precise, automated edits without user-provided inputs like scribbles or bounding boxes. Key innovations include Localization Modulated Image Guidance (LMIG), which integrates text-driven coarse localization with image-driven refinement by modulating cross-attention features in an IP-Adapter model. It provides extensive qualitatively and quantitatively evaluation results.

**Strengths:**

1. **Good design of editing pipeline**: I like the whole design of editing pipeline, where the localization and texture prediction evolves through two branches, and localization is then used in LMIG. Although it still needs some warmup steps in localization, it’s already very different from prior methods which usually adopt two or three separate editing stages.
2. **Novel design of the masked cross-attention**: Although the method is well-grounded on previous techniques, it still introduces smart modifications which needs careful observations. The use of masked cross-attention convincingly demonstrate the necessity of itself, and it’s natural to be included in this framework.
3. **Extensive convincing results**: I think the provided qualitatively results are compelling, with diverse examples across different meshes and references. Also the quantitative metrics and perceptual studies provide strong evidence of superiority over prior methods. And I really appreciate the analysis of robustness to initialization and compositionality.

**Weaknesses:**

1. Although the design of the localization prediction is appreciated, it is limited to text prompt. Do the authors think this framework also supports user-specified mask or localization?
2. **Computational Efficiency**: Optimization requires 4 hours per edit on an A40 GPU, with reasonable results after 1 hour. It lacks comparisons to faster alternatives or discussions on acceleration.
3. **Limited Editing Operations Scale**:  it seems like it only supports texture synthesis on the medium scale part of the object, which can be well described by the text prompt SDS loss. But how about the texture synthesis on parts of smaller scale, like nails on the fingers of a hand and so on? I believe that requires new design of the localization prediction.

**Questions:**

- How sensitive is LMIG to the choice of cross-attention weighting? Are there guidelines for tuning these across different mesh complexities or reference image styles?
- The authors mention about the extensions to NeRFs or videos—could you elaborate on potential challenges, such as view consistency or temporal coherence, in adapting LMIG to these domains?

---

> ### Author Response · Authors · 2025-11-21
>
> We thank the reviewer for their insightful and thorough feedback. We address their comments below.
>
> **W1: User specified localization mask**
>
> While we initially use text alone to obtain a coarse localization, a key strength of our method is its ability to refine this estimate using image guidance so that it conforms to the structure of the object in the reference image. In Fig. 5 we can see this process in action: our method’s vest localization conforms precisely to the shape of the reference vest. In contrast, the “text only” case generates a generic vest localization that does not match the shape of the reference vest. We use text for the initial mask to automate our pipeline. However, our method supports refining arbitrary masks to conform to the reference image. Fig. 10 in the supplemental material (L718 in the PDF) shows our method starting from an initial _user-provided_ mask (heart shaped sunglasses) and uses our image guidance to morph the mask to better fit the object shape in the reference image. In general, arbitrary masks can be provided as an initial guess, and our image guidance will refine them to conform to the object in the reference image.
>
> **W2: Computational efficiency**
>
> While our method requires more computation than some global texturing approaches, no other work directly produces _local_ textures on _meshes_. Thus, our approach is the only and therefore the fastest for this specific task. _Global_ texturing methods have found success accelerating generation by backprojecting multi-view images to a texture map. However, using back projection for local texture generation with explicit localization maps is more difficult. From many views, the localization region is not visible and thus generating multi-view images for localization on these views lacks the context for the local region we are describing.
>
> **W3: Editing operation scale**
>
> The scale of edits is largely tied to visibility across views during our optimization. Larger edits can easily be seen from most views. Smaller edits are not seen from most views which adds noise to the optimization. It is possible to use zoomed in camera parameters and fix the views on a specific smaller region of the mesh to obtain more fine-grained edits. If this is of interest, we can include such an experiment.
>
> **Q1: Cross-attention weighting in LMIG**
>
> For all results shown in the paper, we compute standard cross attention between the entire reference image and our generated image, and then mask these attention values with our continually updated learned localization region from our localization branch. We then add the masked image attention to the standard text-conditioned output to get our final attention hidden states. We experimented with different weights in the sum between text and image attention, but found an equal weighting of both terms to produce the most consistent results over all of our examples. While using a higher weight for the image attention term can often produce local textures that are more closely aligned with the reference image, we found that it does so at the expense of oversaturated colors and less plausible localizations on the input mesh during our distillation.
>
> **Q2: Extension to NeRF and Video**
>
> LMIG relies on being able to predict an explicit localization region on the surface of the mesh. This mask is first rendered to be used in the LMIG attention and then also used to explicitly mask the globally predicted texture on the surface. Implicit representations such as NeRFs tend to prioritize visual coherence of renders and often produce results with subpar geometry (i.e. a collection of floaters that from most views appears like a surface, but is actually disjoint). Thus, representing localization probabilities directly in this implicit representation could produce plausible 2D masks for LMIG, but might struggle with the explicit masking of the global style predictions. For extending LMIG to videos, as reviewer 4fRa points out, a key challenge is ensuring temporal coherence. Since we would not have an explicit localization mask that is guaranteed to be consistent frame to frame in the same way that we do when rendering multiple views from a single 3D representation, enforcing this temporal consistency of the localization in videos is difficult.

---

### Official Review · Reviewer_sZne · 2025-11-01

**Soundness:** 3
**Presentation:** 3
**Contribution:** 2
**Rating:** 4
**Confidence:** 5

**Summary:**

The paper introduces 3D PixBrush, a method for image-driven local editing of 3D meshes. The method allows users to modify specific regions of a 3D object based on a reference image, automatically determining where and how to apply the edits—without requiring any manual input such as scribbles or bounding boxes. The authors demonstrate the effectiveness of 3D PixBrush across a wide range of meshes and reference images.

**Strengths:**

S1. The storyline of this paper is clear and easy to follow.

S2. Editing the local texture of a mesh driven by a reference image is an interesting topic.

S3. The proposed framework is reasonable.

**Weaknesses:**

W1. In section 3.4, the authors claim that the key to their method is the ability to supervise the local texture edits with image guidance. However, the use of SDS with image guidance has already been explored in the literature [1, 2]. Novelty of the method and contributions are limited.

W2. The paper only reports the CLIP score as the quantitative evaluation results, which is not comprehensive for quantitative evaluation. All the ablation studies are demonstrated by qualitative examples. I have no idea whether these results are cherry-picked or not. Moreover, it’s a 3D editing task. The paper only showcases single-view results. There are no multi-view renders or videos of the generated 3D mesh.

W3. In line 305, the authors said, “see supplementary material.” However, I didn’t find the supplementary material.

[1] IPDreamer: Appearance-Controllable 3D Object Generation with Image Prompts.
[2] VP3D: Unleashing 2D Visual Prompt for Text-to-3D Generation.

**Questions:**

Please see the weakness.

---

> ### Author Response · Authors · 2025-11-21
>
> We thank the reviewer for their insightful and thorough feedback. We address their comments below.
>
> **W1: Contribution over existing methods in 3D generation**
>
> While existing methods have explored SDS with image guidance for 3D tasks such as generation, _no existing methods are able to perform image-guided local texturing on meshes_. A naive application of image-guided SDS to the task of local texture editing is shown in Fig. 2, right half, middle column. Observe that this naive approach matches elements from the reference image (i.e. the color of the sunglasses is mostly correct), but does not localize them in the context of the input mesh (the sunglasses are placed on the side of the object instead of on the face). To address this, we introduce LMIG, a technique for using a supervision signal from the reference image to influence two key components of our optimization:
> 1) it learns a mask on the surface that specifies where the local texture should be applied.
> 2) it uses the current predicted state of the mask to control which parts of the supervision image can be influenced by the reference image.
>
> These two contributions enable our method to obtain region-specific guidance from the reference image (see Fig. 2, right half, right column). Specifically, we see that the sunglasses texture is now properly contained within the predicted segmentation region on the front of the face.
>
> **W2 a: Quantitative evaluation beyond the CLIP R-Precession metric**
>
> In addition to the CLIP R-Precision metric reported in the main paper, we include a user perceptual study in the supplemental material (in original PDF, after references, L761) as well as evaluations using CLIP similarity scores (L798) and the LPIPS metric (L805). Additionally, we now include an evaluation using VQA score [1] below. For this metric, we use the same 21 mesh and reference image combinations. The VQA score is computed according to [1] which asks a VQA model the question “Does this figure show \<text\_description\>? Please answer yes or no.” and then outputs the decoder’s predicted probability of “yes.” Our text descriptions are created by using the format “A 3D render of a gray \<object\_name\> with \<BLIP2 caption of reference image\>.” We note that using BLIP2 to caption the reference image often does not perfectly capture all of the subtleties of the image (for example see the BLIP2 generated captions in Fig. 7). However, this only favors the baseline method 3D Paintbrush since their local edits are generated directly from text prompts containing these BLIP2 captions that we evaluate VQA on. Still, our approach scores significantly higher on this VQA metric.
>
> | Method        | VQA Score |
> |---------------|-----------|
> | 3D PaintBrush | 0.48      |
> | **Ours**      | **0.62**  |
>
>
> **W2 b: Multi-view results**
>
> We now include multi-view results in an updated PDF (Fig. 20 in the supplemental, L1070).
>
> **W3: Location of supplemental material**
>
> The supplemental material is located after the references in the main pdf starting on L702. We are also now attaching it separately in OpenReview as its own PDF.
>
> **References:**
> [1] Lin, Zhiqiu, et al. "Evaluating text-to-visual generation with image-to-text generation." European Conference on Computer Vision. Cham: Springer Nature Switzerland, 2024.

---

### Author Response · Authors · 2025-12-03
**Summary for the AC**

Dear AC,

In this work we propose a method for local, image-driven texturing on meshes. A key contribution of our work is our LMIG module. This technique allows us to focus the image supervision so that it is applied only in the localization region and produces a texture that conforms to the shape and context of that region. Reviewers felt that the proposed task of performing both localization and local texturing using a reference image as guidance and without user localization input was both new and interesting (sZne, QuJ6). They comment on the extensive quantitative and qualitative results demonstrating the superiority of our method over baselines (4fRa, o6on, QuJ6) as well as the convincing figures that show our method produces “visually pleasing” results on “diverse examples across different meshes and references” for this new task (QuJ6, 4fRa). Reviewers appreciated the design of our editing pipeline (4fRa) and our clear and well explained method (sZne, QuJ6).

Reviewers had mixed comments on the novelty of our proposed approach. Reviewer 4fRa commented on the novel design of our LMIG module citing its “smart modifications” and necessity for our task. Reviewer o6on points out similarities with 3D Paintbrush and reviewers sZne, o6on reference existing image-guided SDS approaches. **However, no existing method performs image-guided local texturing on meshes**. 3D Paintbrush [1] enables local texturing, but only uses text as input and does not support image guidance. Existing methods for image-guided texturing on meshes have used SDS with IP-Adapter [2], but only for *global texturing*. A trivial combination of IP-Adapater SDS and 3D Paintbrush does not work (see rebuttal response to sZne under “W1”). We solve this problem by introducing our LMIG module which uses our current localization prediction at each iteration to mask the cross attention. See rebuttal responses to sZne and o6on for more details.

Note that reviewers sZne, QuJ6 missed some additional quantitative evaluations in the supplemental (perceptual user study, LPIPs metric), which we clarified in the rebuttal. We additionally add a new quantitative comparison with VQA Score in the rebuttal, further demonstrating the clear improvement of our approach over the baseline. See rebuttal responses to sZne and QuJ6 for more details.

In conclusion, our method is the first to perform image-guided local editing on meshes with explicit localization regions. Key to our method is our LMIG module which allows us to focus the image supervision so that it is applied only in the localization region and produce a texture that conforms to the shape and context of that region. As pointed out by reviewers, our results show convincing improvements over the closest baseline, highlighting the effectiveness of our approach.

References:
[1] Decatur, Dale, et al. "3d paintbrush: Local stylization of 3d shapes with cascaded score distillation." Proceedings of the IEEE/CVF conference on computer vision and pattern recognition. 2024.
[2] Perla, Sai Raj Kishore, et al. ‘EASI-Tex: Edge-Aware Mesh Texturing from Single Image’. ACM Transactions on Graphics (Proceedings of SIGGRAPH), vol. 43, no. 4, ACM New York, NY, USA, 2024, https://doi.org/10.1145/3658222.

---

### Meta-Review · Area_Chair_GZmR · 2026-01-06

**Summary:**

The paper proposes a method for image-driven local texture synthesis of 3D meshes. It leverages a reference image and a text prompt to predict a localization mask and corresponding texture for a 3D mesh automatically. The paper received two 6 ratings (4fRa, QuJ6), one 4 rating (sZne) and one 2 rating (o6on). The reviewers acknowledged that 1) the proposed framework is reasonable and the problem is interesting; 2) the experimental results prove the effectiveness of the approach. The raised critical concerns are mainly on the technical contribution/novelty of the proposed method (sZne, o6on and QuJ6), by considering its incremental extension of 3D PaintBrush (CVPR 2024). The AC also checked the 3D PaintBrush work and agrees with the reviewers.

**Reviewer Concerns:**

The critical concern regarding the novelty/contribution of the method is not fully addressed, as also pointed out by reviewer QuJ6 ("the answers provided by the authors are not satisfactory"), who initially did not raise this concern actually.

**Reviewer Scores:**

The reviewers might not change their ratings, based on the initial comments and rebuttal. Even QuJ6 did not downgrade his score (rating 6), he pointed out he will not argue for accepting the paper.

---

### Decision · Program_Chairs · 2026-01-26

Reject